Flood analysis comparison with probability density functions and a stochastic weather generator

García-Ledesma Israel
Madrigal Jaime jose.madrigal@umich.mx
Pardo-Loaiza Jesús
Hernández-Bedolla Joel
Domínguez-Sánchez Constantino dsanchez@umich.mx
Sánchez-Quispe Sonia Tatiana
Faculty of Civil Engineering, Universidad Michoacana de San Nicolás de Hidalgo , Morelia , Michoacán , Mexico
Fu Guobin
Electronic publication date: 2025 May 5
Publication date: 2025
Volume: 13
Electronic Location ID: e19333
Received 2024 Jul 11; Accepted 2025 Mar 26
Copyright: ©2025 García-Ledesma et al.
Copyright year: 2025
Copyright holder: García-Ledesma et al.
License: This is an open access article distributed under the terms of the Creative Commons Attribution License, which permits unrestricted use, distribution, reproduction and adaptation in any medium and for any purpose provided that it is properly attributed. For attribution, the original author(s), title, publication source (PeerJ) and either DOI or URL of the article must be cited.
License URL: https://creativecommons.org/licenses/by/4.0/

Keywords: Flood risk management, Urban planning, Stochastic weather generator, Hydrodynamic modeling, Flood forecasting

Funding: Ministry of Science, Humanities, Technology, and Innovation (SECIHTI - Secretaría de Ciencia, Humanidades, Tecnología e Innovación) Israel García-Ledesma and Jaime Madrigal received funding from the Ministry of Science, Humanities, Technology, and Innovation (SECIHTI - Secretaría de Ciencia, Humanidades, Tecnología e Innovación) for their postgraduate studies. The funders had no role in study design, data collection and analysis, decision to publish, or preparation of the manuscript.

==============================
Flood prediction has become essential to hydrology and natural disaster management due to the increasing frequency and severity of extreme hydrological events driven by climate change. This study compares two methodologies for predicting flood events in Morelia, Mexico: theoretical distribution functions and stochastic weather generators. The methodology integrates maximum runoff results for different return periods into a drainage network hydraulic model, using the Soil Conservation Service Curve Number (SCS-CN) method and a multivariate stochastic model (MASVC). Hydrodynamic modeling with HEC-RAS, incorporating two-dimensional shallow water equations, was used to simulate flood inundation areas. The study reveals that while both modeling approaches similarly replicate the system’s behavior, they produce different water levels due to variations in maximum flow values. The stochastic model tends to generate higher maximum water levels. High-resolution digital elevation models (DEMs) with a pixel size of five m in urban areas and 0.5 m in drainage network zones, and land use data were crucial in improving the accuracy of the hydraulic simulations. Findings indicate that unregulated urban growth in flood-prone areas significantly exacerbates the impact of flooding. The generated hazard maps and flood simulations provide valuable tools for urban planning and decision-making, highlighting the need for strategic interventions to mitigate flood risks. This research underscores the importance of integrating advanced modeling techniques in flood risk management to enhance the precision and reliability of flood predictions.

Introduction

In recent decades, flood forecasting and prediction have become critical to hydrology and natural disaster management. The increasing frequency and severity of extreme hydrological events, primarily due to climate change, highlight the need to enhance the precision and reliability of prediction systems. The ability to accurately and timely prediction flood events is essential to reduce property loss, protect vital infrastructure, and prevent human casualties (Papaioannou et al., 2021).

Flood prediction is a multidisciplinary field combining information from meteorology, hydrology, geography, and computer science, among other disciplines. Progress in these areas has led to more advanced prediction models and methods. However, natural systems have variability and uncertainty, and human–environmental interactions are complex. These significant challenges have not yet been solved (Chahinian et al., 2023).

Adopting emerging technologies such as artificial intelligence, machine learning, and big data has opened new ways for flood risk analysis and modeling (Karyotis et al., 2019; Mosavi, Ozturk & Chau, 2018). These approaches promise improvements in the ability to predict floods by enabling the processing of large volumes of data and the identification of complex patterns imperceptible to traditional methods (Falconer et al., 2009). In addition, the increasing availability of high-resolution data obtained through satellites and other remote sensing offers unprecedented opportunities to improve flood prediction accuracy and spatial resolution (Munawar, Hammad & Waller, 2022).

Methods based on Machine Learning are, without a doubt, among the most analyzed in recent years. This type of methodology includes adaptive neuro-fuzzy inference systems (ANFIS), multilayer perceptron (MLP), artificial neural networks (ANNs), wavelet neural networks (WNN), and support vector machines (SVM), among others.

Choubin et al. (2016) analyzed precipitation prediction using three models: multiple linear regression (MLR), MLP, and ANFIS, using large-scale climate signals as inputs. Effective climate indices were selected through principal component analysis and cross-correlation to predict the standardized precipitation index (SPI) in the Maharlu-Bakhtegan basin, Iran. The results indicated that the MLP model outperformed the MLR and ANFIS models, suggesting a nonlinear relationship between climate signals and precipitation, making nonlinear methods more effective for predicting the analyzed area.

Gessang & Lasminto (2020) propose using ANN and a weather forecasting API for flood prediction and mitigation in a sub-basin of Indonesia. The research highlights how the precipitation intensity in this area can cause significant increases in river water levels, leading to flooding. ANN was used to predict rainfall and, together with the curve method (CN) of the United States Soil Conservation Service (USSCS), calculate maximum runoff, demonstrating the usefulness of these models in flood risk management.

Hernández-Bedolla et al. (2023) present a multivariate and multisite stochastic model, MASVC, designed to estimate maximum runoff in non-measured basins. The study highlights how precipitation influences the determination of runoff at different time scales and uses a stochastic approach to generate synthetic precipitation sequences, preserving spatial and temporal variability in daily, monthly, annual, and extreme values. The model was evaluated in the Rio Grande watershed of Morelia, Mexico. It showed its effectiveness by contrasting its results with conventional probability density functions and providing a more dependable approximation of peak surface runoff.

Several factors prevent the successful application of various flood analysis and prediction technologies in operational practice despite the numerous methods that exist for this purpose. These include the need for comprehensive and accurate historical datasets, understanding local river and rainfall dynamics, and integrating heterogeneous models and data into cohesive and reliable operating systems (Perera et al., 2020).

The most commonly used flood prediction methods use maximum precipitation associated with theoretical distribution functions as input. However, rainfall values—and consequently flow rates—for significant return periods exceeding 50 years tend to exhibit uniform behavior. This behavior could indicate a potential systematic bias in the projected results. For this reason, the present study proposes an alternative for generating synthetic flows using a stochastic model. Once the stochastic flows are obtained, a hydraulic modeling of the main rivers in Morelia, Mexico, is carried out. This comparison aims to assess an alternative option to the current methods for predicting and evaluating urban flood events.

Materials & Methods

Case study

Morelia, the largest and most populous urban area in Michoacán, Mexico, is characterized by its significant population and complex fluvial network. The city has a fluvial network of fifteen perennial tributaries, as shown in Fig. 1, and is the object of analysis in this work. The Rio Grande is the city’s main river channel, collects the runoff generated in the area, and is regulated upstream by the Cointzio dam.

Figure 1 River network of the city of Morelia.

Created using QGIS 3.18.

Morelia is prone to frequent floods from overflowing rivers, which directly impact houses and disrupt residents’ daily lives. For instance, in 2018 a severe storm caused injuries, inundations in homes, landslides, and wrecked vehicles as shown in Fig. 2. Today, floods are perceived as having only negative consequences and as disasters that inflict harm, sometimes irreparable. These impacts are aggravated by the expansion of cities that are naturally prone to flooding, such as near rivers and streams.

Figure 2 Photographs of the flood in Morelia in 2018.

Methodology

The proposed methodology utilizes maximum runoff results for various return periods within the hydraulic model of Morelia City’s drainage network. It also incorporates scenarios created with the daily MASVC (Hernández-Bedolla et al., 2023). Moreover, the HEC-RAS model was adjusted to obtain the flood areas based on topographic studies and changes in the Manning roughness coefficients (Demir & Keskin, 2020; Yalcin, 2020). The methodology is illustrated in Fig. 3.

Figure 3 Proposed methodology.

Surface runoff

Surface runoff was calculated using two types of methodology. One method was the probability density functions with the Soil Conservation Service Curve Number (PDF-SCS-CN), which is used to estimate runoff from small-to-medium-sized watersheds SCS-CN method (AL-Hussein et al., 2022; Ansori, 2023; Sathya, Thampi & Chithra, 2023). The other method was the stochastic weather generator, which applied the MASVC-SCS-CN approach (Hernández-Bedolla et al., 2023). The MASVC is a multivariate stochastic model that uses lag one autoregressive multivariate parameters. It consists of two modeling phases. The first phase models the occurrence of precipitation (wet-dry), and the second phase estimates the amount of precipitation on a daily scale. Then, the maximum precipitation values are extracted, and the runoffs are computed using the SCS-CN method. The observed daily precipitation was obtained from the National Water Commission (CONAGUA). The weather stations used were 16,022, 16,055, 16,114, and 16,247 (Fig. 1). The historical period of record was from 1980 to 2009, and it is available for the four rainfall stations (https://smn.conagua.gob.mx). In addition, hourly precipitation for October 22, 2018, from the Automated Weather Stations (AWS) database, available at https://smn.conagua.gob.mx/es/observando-el-tiempo/estaciones-meteorologicas-automaticas-ema-s, was used. The hydrographs for the modeling were derived from the maximum flows suggested by Hernández-Bedolla et al. (2023).

Flood inundation

Hydrodynamic model HEC-RAS is used to model rivers (one-dimensional), flood areas (two-dimensional), channels, drains, and dams (Açıl et al., 2023; Bharath et al., 2021; Bush et al., 2022; Goswami, Prasad & Kumar, 2023; Namara, Damisse & Tufa, 2022; Ongdas et al., 2020). HEC-RAS calculates the depth of inundation and velocity based on floodwater discharge hydrographs (Hydrologic Engineering Center, 2009). HEC-RAS was applied for the two-dimensional hydraulic model and was solved by the 2-D shallow water equations. The solution method for both equations was by volume finite differences. The unsteady differential form of the mass conservation equation is Eq. (1). (1) r=∂H∂t+∂hu∂x+∂hv∂y

where r is the source/sink term, H is the surface elevation (m); t is the time, hu and hv are the flow in x and y (m2s−1); HEC-RAS recently incorporates fully 2-D shallow equations (Costabile et al., 2020). The shallow water equations are presented in Eqs. (2) and (3).

(2) ∂p∂t+∂∂xp2h+∂∂ypqh=−n2pgp2+q2h2−gh∂H∂x+pf+∂ρ∂xhτxx+∂ρ∂yhτxy

(3) ∂q∂t+∂∂xpqh+∂∂yq2h=−n2pgp2+q2h2−gh∂H∂y+qf+∂ρ∂xhτxy+∂ρ∂yhτyy

where p and q are hu and hv, (m2s−1); n is the Manning’s roughness coefficient (s m−1/3), g is the gravity acceleration (ms−2), ρ is the water density (kg m−3), τxx, τyy, and τxy are the components of the stress tensor and f is the Coriolis parameter (s−1). We propose the diffusive wave algorithm to select boundary conditions and response times. In this case, the inertial terms of the equation are neglected.

The Eulerian Shallow Water equation (EM-SWE) was the calibration solution method. This method utilizes the momentum-conservative discretization assuming local conservation of momentum about control volume centered on all cell face V⋅∇uN (Hydrologic Engineering Center, 2021). (4a) V⋅∇uNf≈1h∇⋅hVuN−uN∇⋅hVf

(4b) V⋅∇uNf≈αfLh¯fAL ∑k∈KLQi,k−Vku⋅nf−uN,f+αfRh¯fAR ∑k∈KLQR,k−Vku⋅nf−uN,f

where hi=Ωi/Aiw; Qi,k__ is the minimum value of inflow at face k to cell i si,k, Qk, 0;  Ak is the face vertical area; AL is the left cell horizontal area, AR is the right cell horizontal area, Vj is the cell average current velocity vector Vku is VL for uN,k major than cero and is VR for uN,k less than zero; nf is the face-normal unit vector.

DEM and land use

The quality of flood modeling depends on the topographic information. A hybrid mesh was used, combining a gridded DEM dataset with five m of resolution from the National Institute of Statistics and Geography (INEGI, https://www.inegi.org.mx/temas/topografia/) and more detailed 0.5 m data from a specific topographic study to improve the accuracy in rivers. The land use data was obtained from INEGI (https://www.inegi.org.mx/temas/usosuelo/), and field visits to the rivers were conducted to determine the Manning coefficient (n).

Boundary conditions

Two-dimensional modeling was done with flexible and triangular meshes. The meshes gave more detail in the rivers due to a high-accuracy topographic survey (Muñoz et al., 2022; Ontowirjo et al., 2023). For the flood areas, a regular grid with a 5-meter resolution was proposed for roads, as well as for urban and rural areas. Boundary conditions describe how water behaves at the model domain’s boundaries, including different conditions at the model edges (Hydrologic Engineering Center, 2009). Upstream boundary conditions are needed at the upstream end of all reaches that do not connect to other reaches or storage areas. Hydrographs corresponding to various return and historical periods were assigned upstream of each sub-basin studied. For the downstream boundary condition, the normal depth option was chosen for the HEC-RAS model (Hydrologic Engineering Center, 2021).

Simulation and calibration

The process of calibrating the hydraulic model is based on the different steps described below. (1) Simulation of the entire study area to identify the simulation times and potential flooding zones near the streams and rivers. (2) Simulation of the diffusive wave algorithm to generate the first flood from historical rainfall data. (3) Calibration using the EM-SWE by changing the Manning roughness coefficient from 0.01 to 0.75, and field visits to perform mesh refinement. (4) Validation of the results from the actual flooding for a specific date. In this case, validation was conducted by juxtaposing the simulation results with an orthophoto that documented the flooding event which occurred on October 22, 2018 (Roblero-Escobar et al., 2025). (5) Generation of floods from different return periods and results from MASVC-SCS-CN and PDF-SCS-CN. For the MASVC, we generated 1,000 series of the same length as the observed rainfall. For PDF, we consider the annual maximum rainfall. Subsequently, we estimated the surface runoff based on curve number “CN” and calibrated de hydraulic model changing n’s Manning.

Results and Discussion

The city of Morelia often experiences floods due to the excess water in the urban drains that are part of the city’s structure. In order to analyze the flooding problem accurately, a high-resolution (five m) digital elevation model (DEM) of the city is needed. This study used nine DEMs provided by INEGI (Instituto Nacional de Estadística y Geografía (INEGI), 2024) to create a digital surface model (DSM) that reflects the city’s structure. This DSM allowed us to locate the potential channels that could form when the drains overflow (Table 1).

Table 1 River Network of the city of Morelia.

River network of the city of Morelia	
Name	ID	Channel length (km)	Total ΔH (m)	Average S (%)	
R. Grande	01	29.35	12.144	0.04	
Alberca	02	2.41	4.34	0.18	
Calabocito	03	1.53	2.63	0.17	
Calabozo	04	4.90	106.74	2.18	
Itzícuaros	05	4.40	0.62	0.01	
Parían	06	2.90	12.01	0.41	
Barajas	07	3.78	10.35	0.27	
Arroyo Blanco	08	2.96	37.69	1.27	
Arroyo de Tierras	09	9.09	246.12	2.71	
Mora Tovar	10	1.69	8.46	0.50	
Río Chiquito	11	9.24	104.11	1.13	
Carlos Salazar	12	0.57	17.54	3.06	
Soledad	13	4.03	1.16	0.03	
Quinceo	14	6.60	9.00	0.14	
Erandeni	15	3.52	62.16	1.77	

However, the DSM has a limitation in that it does not account for vegetation. Vegetation can lower the hydraulic capacity of waterways, raising the flood risk. Bathymetric data on the river network was obtained from CONAGUA (2016) to obtain a comprehensive topographic model of the urban area. This data was combined with the DEM of the urban layout to create a complete topographic model. The final topographic model enabled the precise identification of the city’s location, depth, and areas prone to flooding.

The hydraulic simulation was performed with the HEC-RAS version 6.3 program. This software applies the two-dimensional shallow-water equations of diffusion waves and lets users select the best solution method.

Shallow water equations are a set of mathematical equations that capture the movement of water in open channels. These equations can simulate different hydraulic phenomena such as floods, drinking water flows, and wastewater, which is why they were applied in this study.

The hydraulic model covers a 34.74 km2 area, split into 751,641 cells and 871,976 nodes. The cells have varied sizes depending on the area, with large-scale cells in areas away from the channels (10 m) and small cells in the channels (two m) and structures (one m). This variable size enhances the calculation process without losing detail in critical areas. The model includes 170 structures, such as bridges, gates, and culverts. These structures reflect the actual topography of the area and allow the water flow around them to be modeled.

The hydraulic model requires 17 boundary conditions. Of these, 16 are water inlets, represented by the hydrographs of each tributary, and one is an outlet located in the final section of the Rio Grande outside the city. This configuration enables the flow within the study area to act naturally, conforming to the terrain’s topography. Figure 4 shows the position of boundary conditions in the model.

Figure 4 Boundary conditions for the hydraulic model.

Created using QGIS 3.18.

A comparative analysis was used to obtain the hydrographs for the water inlets. Two methods were applied: the best-fit probability function (Sánchez-Quispe, Navarro-Farfán & García-Romero, 2021) and stochastic models (Wright, Yu & England, 2020). Both methods were used to estimate design storms.

The hydrographs for the modeling were derived from the maximum flows suggested by Hernández-Bedolla et al. (2023), using precipitation information from the period between 1980 and 2009. The model input hydrographs, 15 in total, were produced using the HEC-HMS software. The constant flow of the Cointzio dam was based on its historical discharges.

Figure 5 shows that the use of stochastic models results in consistent maximum flows, regardless of the methodology (PDF-SCS-CN or MASVC-SCS-CN) used, because these models are uniform over time and reduce the uncertainty caused by the significant variation of using a best-fit distribution function, which changes between basins and seasons. In addition, since the headwater (non-urban) micro basins have high precipitation, the adjustment function is the log-Gumbel. In contrast, the log-normal distribution function (lower precipitation) was adjusted in the urban micro basins.

Figure 5 Peak runoff flows resulting from the precipitation events in the microbasins of the study, MASVC-SCS-CN, and PDF-SCS-CN generated in Hec-HMS.

(A) Itzicuaros drain, (B) Quinceo drain, (C) Chiquito river, and (D) Arroyo de tierras drain.

Manning’s roughness coefficient is critical for hydraulic simulation as it influences the flow speed. For instance, higher roughness coefficients typically indicate more resistance and slower flow velocities. Satellite images were used as a basis for this. Chow (1959) states that this coefficient can be derived from land cover. However, the Manning coefficients were verified to ensure the parameter’s accuracy in the system’s hydraulic performance. This involved field visits and topographic surveys.

The data gathered enabled the identification of areas with comparable features of land cover and soil type. Each zone was given a distinct roughness coefficient. Figure 6 illustrates the spatial variation of these coefficients.

Figure 6 Manning’s roughness coefficients per zone.

Created using QGIS 3.18.

Figure 7 Natural floodplains of the river network.

Created using QGIS 3.18. Zone A, corresponding to the Alberca Drain, experiences flooding with a water depth of 1.0 meters. Zone B, associated with the Itzicuaros Drain, is the most affected area, showing the largest flooded zone and the greatest depth, with water levels reaching up to 2.5 meters. Zone C is affected by flooding due to the overflow of the Río Grande. Meanwhile, Zone D, which collects water from the Soledad and Quinceo drains, represents the second most affected area, with flood depths reaching up to 1.0 meters.

Figure 8 Hydraulic simulation for 100-year return period, where flood zones are shown.

Created using QGIS 3.18.

Different return periods (2, 10, 50, 100, 200, and 500 years) were used for hydraulic simulations. These simulations revealed four zones of natural flooding, as shown in Fig. 7, which presents the results for a 100-year return period. This information is essential for deciding the city’s urban growth, as these areas are prone to flooding.

The input data inevitably creates variation in the depths produced on the flood surfaces depending on the model used, but the flood areas are clearly defined, as shown in Fig. 8. This variation in the generated depths allows it to remain safe without significantly affecting the flood surface, as the hydraulic response of the system in the study area is perfectly defined.

As shown in Fig. 8, regardless of the input data, the hydraulic response of the area is similar, with variations occurring only in the depths. The differences are due to the maximum discharge values generated by each model. Generally, the variations in the flood zones observed in Figs. 7A, 7B, 7C, and 7D are more significant in the stochastic models, as seen in Table 2.

The selection of an appropriate distribution is static and based on yearly maximum rainfall (one data per year). In the PDF, we use the error in parameter estimation.

The confidence intervals are used for the parameters of the distribution; the error in the present paper is the 5% (p ≤ 0.087). On the other hand, MASCV involves a dynamic and complex approach that can capture temporal and spatial evolution with smaller uncertainty (1,000 series of 365 days). The mean, standard deviation, skewness coefficient, and lag-one autocorrelation are calculated with 95% confidence. Moreover, the MASCV evaluated observed and generated series do not follow the K–S, t-test, and Wilcoxon (Hernández-Bedolla et al., 2023). The maximum rainfall oscillates ± 15 mm for all 1,000 series.

Conclusions

All cities, particularly those prone to natural hazards like flooding, must address the challenges posed by unregulated urban growth. By occupying natural flood areas, economic and social losses pose a severe threat, which becomes a vital issue for governments.

The findings of this study highlight the necessity of using hydraulic models as essential tools in urban planning and decision-making to mitigate impacts on existing urban centers. Assessing natural hazards and their effects, such as rainfall that causes floods, is a complicated task that requires rigorous analysis, as done in this research. Although an uncertainty analysis is not performed directly, it is implicit in this article. By analyzing rainfall on a daily scale, extreme events and the seasonal and interannual variability of precipitation are captured. This more significant amount of information allows us to obtain a more robust estimate of the parameters of the distribution of extreme precipitation. In the case of analyzing the PDF’s-SCS-CN-HEC-RAS of annual maximum precipitation, only a small portion of the available data is being analyzed, which may result in a less accurate estimation of the probability distribution. Therefore, MASVC-SCS-CN-HEC-RAS decreased uncertainty in floods and areas with limited data, giving more confidence to the decisions that need to be made.

Table 2 Maximum depth level reached in flooding areas.

Method	Tr	Average maximum depth (m)	
		a	b	c	d	
Distribution f.	2	0.35	0.32			
Stochastic	0.37	0.32			
Distribution f.	10	0.43	0.95		0.18	
Stochastic	0.38	0.85	0.21	0.19	
Distribution f.	50	0.56	1.37	0.69	0.23	
Stochastic	0.57	1.69	0.74	0.29	
Distribution f.	100	0.65	1.62	0.73	0.29	
Stochastic	0.68	2.01	0.91	0.38	
Distribution f.	200	0.77	2.02	1.15	0.35	
Stochastic	0.68	2.01	0.91	0.38	
Distribution f.	500	1.06	2.18	1.76	0.42	
Stochastic	1.53	2.78	2.11	0.64	

A crucial task in decision-making is the generation of MASVC-SCS-CN-HEC-RAS models that simulate the flow behavior. Therefore, it is essential to be careful when creating the model. Verifying every aspect involved in the rainfall-runoff and hydraulic model is essential. Two-dimensional hydraulic models accurately simulate the flow behavior as it happens. The generation of hazard maps, such as those for flooding, helps to identify the areas where urban development should be avoided because they are in areas prone to flooding, such as the natural floodplains of the river network of Morelia.

Flood risk analysis can be addressed using two accepted methodologies: flow rates obtained through best-fit distribution functions and stochastic methods. Both techniques have their merits and are widely used in hydraulic evaluation. However, it has been observed that flow rates determined by stochastic means tend to underestimate values during the first 30 years of analysis, and subsequently, they may overestimate them. It is crucial to consider these variations when selecting the appropriate methodology for analysis, ensuring precision and reliability in flood risk management.

The hazard maps produced in this study can help to identify strategic locations for hydraulic works that can help to mitigate the impacts of flooding in the urban area of Morelia. By analyzing different return periods, the maps clearly show the recurrent flood zones that should be prioritized for intervention. The flood maps can also be helpful for future work in developing warning systems that can link rainfall events with the historical floods generated.

The authors thank the Universidad Michoacana de San Nicolás de Hidalgo (UMSNH) for providing the necessary study facilities.

Additional Information and Declarations

Competing Interests

Author Contributions

Data Availability

The authors declare there are no competing interests.

Israel García-Ledesma performed the experiments, analyzed the data, prepared figures and/or tables, and approved the final draft.

Jaime Madrigal conceived and designed the experiments, analyzed the data, authored or reviewed drafts of the article, and approved the final draft.

Jesús Pardo-Loaiza performed the experiments, analyzed the data, prepared figures and/or tables, and approved the final draft.

Joel Hernández-Bedolla performed the experiments, prepared figures and/or tables, and approved the final draft.

Constantino Domínguez-Sánchez conceived and designed the experiments, authored or reviewed drafts of the article, and approved the final draft.

Sonia Tatiana Sánchez-Quispe conceived and designed the experiments, authored or reviewed drafts of the article, and approved the final draft.

The following information was supplied regarding data availability:

The topography data is available at INEGI: https://en.www.inegi.org.mx/temas/topografia.

The use of soil and vegetation data available at INEGI: https://en.www.inegi.org.mx/temas/usosuelo.

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
