# Peer review of "Flood analysis comparison with probability density functions and a stochastic weather generator"

_PeerJ, doi:10.7717/peerj.19333_

## Round 0.1 · original submission · Major Revisions

Your manuscript has been reviewed by three experts in this field: one recommended a rejection and two minor revisions. I have read the comments and found the minor revision reviewers have a lot of questions and the comments from the reject reviewer (R2) may be addressed if you want. Therefore, I am asking for a major revision.

Reviewer 1 ·

Basic reporting

1. Why did the author select the Morelia, Mexico study area for analysis?
2. Have you considered LULC in your study?
3. What Manning's N value is taken in your study?
4. Which DEM is used?
5. Length of the data?
6. In the abstract, the author mentioned the word "high resolution" - What do you mean by high resolution? Kindly justify it.
7. Which stochastic method was used?
8. Expand the introduction section by highlighting the innovativeness and novelty.
9. Which version of HEC-RAS was used in the study?
10. Equations 2 & 3 are not cited in the text.
11. What boundary conditions have the authors taken in your study for analysis?
12. The conclusion is vague. Re-write the same.
13. Put some flood site visit photos in the study area section.
14. What assumptions authors have considered in their study?
15. What is the internal between one cross-section to another?
16. Give citations wherever required.

Experimental design

No comments

Validity of the findings

No comments

Additional comments

No comments

Reviewer 2 ·

Basic reporting

The manuscript under consideration does not meet the necessary standards for publication. It is very brief and poorly developed, with most critical aspects of the research inadequately explored. Key sections such as the literature review, methodology, results, and discussion are significantly underdeveloped, lacking the depth and detail needed to support the research's claims. Furthermore, the paper fails to clearly explain important concepts and terminologies, leaving readers without a proper understanding of the study's context, objectives, and significance. The data provided is minimal and lacks rigorous analysis, making it difficult to assess the validity of the findings. Additionally, the manuscript is poorly organized, with disjointed ideas and abrupt transitions between sections, making it challenging to follow the author's line of reasoning. The paper also lacks a critical discussion of the results, with no consideration of the implications, limitations, or potential areas for future research. Given the extensive revisions required, it is recommended that the paper be rejected. The authors may need to reconsider their approach, significantly expand their research, and thoroughly revise the manuscript before resubmitting.

Experimental design

no comment

Validity of the findings

no comment

Additional comments

no comment

·

Basic reporting

The report is a good read, but I think it would be very important to ensure that definitions are consistent:

- I would clarify exactly what is meant by flood forcasting (e.g. L. 20, L. 46, L.62, L. 93). The text sometimes uses forecast and sometimes prediction; as I understand it, forecasting in this work refers to a statistical evaluation, not a real-time forecast. In my opinion, this is rather prediction. Either way, I would start by defining more precisely what is meant here, because there are many different ways of looking at it.

- L272: "return periods" I would use a consistent term: return periods or recurrence intervals. I think that the first term is the most widely used (e.g. "recurrence intervals" in abstract).

The following sentence in the introduction is not quite clear to me.

- L88/89: "Despite the progress of the methods that predict flood events using maximum rainfall from
89 distribution functions as an input, these are still the most widely used methods." This sentence should be reworded. What are the most widely used methods?

Experimental design

Unfortunately, the methodology is not completely comprehensible to me. It would be very important for the following points to be addressed.

- It would be helpful if the available data could be explained as a sub-chapter of "Materials & Methods". Unfortunately, it is not clear to me which data was available and used for the paper, especially regarding the climate data. Also looking at Figure 2: "Obtaining historical climate data" What time span and temporal resolution was recorded? Can return periods equal and greater than 100 years be reliably derived from this?

Also regarding the calibration and validation I have some open questions:

- Subchapter "Simulation and calibration": Which events are used for calibration and which for validation? In which value ranges the Manning coefficients are varied? What is the criterion for stopping the calibration process? Specific number of iterations? Which data is used for calibration and validation?

- Figure 2: "Validate inundation" That is not clear for me. What criteria are used to decide whether another iteration follows or not?

Some minor questions I have regarding the methodology are:

- L187: Do I understand it right, that you assume the same return period for the runoff as for the rainfall?

- L240: Why were these return periods chosen? Are e.g. 500 years still representative?

- Figure 6: Which return period is shown here? Which method is used? Log-normal, Log-gumbel or MASVC-SCS-CN?

- Figure 7: Which method is used? Log-normal, Log-gumbel or MASVC-SCS-CN?

- What does "log-gumbel" mean? I assume it is the gumbel distribution shown in log plot. It might be worth showing the equations for the distribution functions in the methodology as well.

Validity of the findings

Some conclusions are incomprehensible to me and need a little more explanation:

- L243-244: Different return periods/recurrence intervals and not different input data are shown. It would be good if the return periods were indicated in the figures.

- L245/246: "It keeps us safe since the hydraulic behavior of the system in the study area is well-defined." What is meant by this? What is the “it” that keeps "us" safe and who is "us"?

- L247-251: What are the errors and uncertainties associated with both methods? Have different realizations been considered in the stochastic models? How much do the results vary with diffrent realizations?

- L260-262: How does this result in a reduction of uncertainties? In my view, uncertainties were not directly taken into account in this study.

- Figure 4: What are the differences in the individual "microbasins" that the log-gumbel is sometimes significantly lower and sometimes significantly higher than the log-normal distribution? In my opinion, this should be discussed in the text.

- Table 2: what does a, b, c and d mean? If this are the microbasins from before, still it should be repeated in the caption. Does "Distribution f." represents Log-normal or gumbel?

Additional comments

I noticed a few small things in the text that should be fixed:

- L267: "helps identify" --> helps to identify

- L271: "help mitigate" --> help to mitigate

- L200: "MDE"

- L 99: "[...] and the object of analysis in this work." The sentence must be adjusted, e.g. "and is the object of analysis in this work."

- L175-177: Sentence is repeated.

---

## Round 0.2 · Minor Revisions

I have read your responses to review comments, which I think you have addressed comments well. However, I cannot find some changes in the manuscript that would address the reviewers' comments. In the response letter, it was not specified if and where the changes were made in the manuscript. For example, Reviewer 3 had several questions/comments about the methods and figures that appeared not addressed in the manuscript. It might be a good idea to modify the manuscript
Look forward to receiving your revised manuscripts.

Reviewer 1 ·

Basic reporting

Ok

Experimental design

Ok

Validity of the findings

Ok

Additional comments

Paper is improved by the authors.

---

## Round 0.3 · accepted · Accept

Thanks for your revision and I think authors have addressed all of the reviewers' comments.